# Fieldwork-based determination of design priorities for point-of-use drinking water quality sensors for use in resource-limited environments

**Michael S. Bono, Jr.** [1,2]*, **Sydney Beasley** [2,3,4], **Emily Hanhauser** [1,2], **A. John Hart** [1], **Rohit Karnik** [1], **Chintan Vaishnav** [2,5]*

**1** Department of Mechanical Engineering, Massachusetts Institute of Technology, Cambridge, MA, United States of America, **2** Tata Center for Technology and Design, Massachusetts Institute of Technology, Cambridge, MA, United States of America, **3** Technology and Policy Program, Massachusetts Institute of Technology, Cambridge, MA, United States of America, **4** Department of Urban Studies and Planning, Massachusetts Institute of Technology, Cambridge, MA, United States of America, **5** Sloan School of Management, Massachusetts Institute of Technology, Cambridge, MA, United States of America

* msb294@cornell.edu (MB); chintanv@mit.edu (CV)

**Data Availability Statement:** All relevant data are within the paper and its Supporting Information files.

## Abstract

Improved capabilities in microfluidics, electrochemistry, and portable assays have resulted in the development of a wide range of point-of-use sensors intended for environmental, medical, and agricultural applications in resource-limited environments of developing countries. However, these devices are frequently developed without direct interaction with their often-remote intended user base, creating the potential for a disconnect between users' actual needs and those perceived by sensor developers. As different analytical techniques have inherent strengths and limitations, effective measurement solution development requires determination of desired sensor attributes early in the development process. In this work, we present our findings on design priorities for point-of-use microbial water sensors based on fieldwork in rural India, as well as a guide to fieldwork methodologies for determining desired sensor attributes. We utilized group design workshops for initial identification of design priorities, and then conducted choice-based conjoint analysis interviews for quantification of user preferences among these priorities. We found the highest user preference for integrated reporting of contaminant concentration and recommended actions, as well as significant preferences for mostly reusable sensor architectures, same-day results, and combined ingredients. These findings serve as a framework for future microbial sensor development and a guide for fieldwork-based understanding of user needs.

## Introduction

Recent advances in microfluidics, electrochemistry, and portable assays have enabled widespread research and development of sensors for point-of-use environmental and health

**Funding:** This research was supported by the Tata Center for Technology and Design at MIT (https://tatacenter.mit.edu/), which was funded with generous support from the Tata Trusts. Travel for EH was funded by the Abdul Latif Jameel Water and Food Security Lab (J-WAFS) at MIT (https://jwafs.mit.edu/), which was funded with generous support from Mohammed Abdul Latif Jameel. Both Tata Center and J-WAFS funding was awarded to CV, RK, and AJH. Employees of the Tata Trusts assisted with arranging travel logistics and stakeholder engagement activities, and employees of the Himmotthan Society, an associate organization of Tata Trusts, assisted with recruitment and translation for the conjoint analysis interviews. Other than the aforementioned assistance, the funders had no role in data collection, and the funders had no role in study design, data analysis, decision to publish, or preparation of the manuscript. The views and conclusions expressed in this article are solely those of the authors, and do not represent the views of the funders for this work.

**Competing interests:** I have read the journal's policy and the authors of this manuscript have the following competing interests: The authors have submitted a utility patent application for potential sensor designs based on the results of this study: Bono MS, Beasley SB, Hanhauser EB, Vaishnav C, Hart AJ, Karnik RN. Systems, Devices, and Methods for Point-of-Use Testing for Fluid Contamination; Filed Oct. 18, 2017. U.S. PCT Patent Application No. PCT/US2017/057265. At the time of article submission, none of the sensor designs described in this patent application have been commercialized, nor are there current agreements in place to commercialize them. This does not alter our adherence to PLOS ONE policies on sharing data and materials.

sensing in resource-limited areas, resulting in improved ability to detect analytes such as ions [1–3], bacteria [4–10], viruses [11], and nucleic acids [12–14]. In particular, there is a strong interest in developing improved point-of-use sensors for detecting contamination in drinking water in resource-limited settings [6, 15] as part of comprehensive water quality management solutions for the 663 million people who currently lack improved drinking water sources and 2.4 billion people total who currently lack improved sanitation facilities [16].

Although water contamination can take the form of physical, chemical, or biological contamination, biological contamination due to the presence of pathogenic microbes is of particular interest for point-of-use sensing due to its widespread occurrence in resource-limited environments and amenability to relatively inexpensive remediation at the household or community level. Unsafe water and lack of sanitation cause 88% of diarrhea cases worldwide [17], and diarrhea in turn causes over 10% of all global deaths for children under 5 [18].

Microbial water contamination may take the form of a wide range of pathogenic bacteria, viruses, helminths, and protozoa [19]. However, microbial water contamination is generally monitored by detecting fecal indicator bacteria such as *E. coli* or thermotolerant coliform bacteria in order to determine if water has come in contact with human or animal fecal matter. The World Health Organization (WHO) has specified a guideline of no culture-forming units (CFU) of indicator bacteria detected in a 100 mL sample for drinking water [15, 19]. In rural resource-limited environments, microbial contamination is frequently remediated at the household or community level via water treatment or improved sanitation. As these methods are dependent on actions by rural end users, behavior change interventions are a critical component of many microbial water contamination remediation programs [20, 21]. As many behavior change interventions involve conveying information that contrasts the current understanding and behavior with those desired, via methods such as awareness campaigns [20, 22, 23], these interventions are also referred to as behavior change communication. These information-based behavior change interventions can be made more effective by providing end users with information about the contamination in their water [24–27]. Such information can be provided in the form of a report from lab-based testing, point-of-use testing performed in the presence of end users by trained personnel, or by providing end users with their own point-of-use tests to use at their discretion [27].

The current commercial state-of-the art for inexpensive point-of-use bacterial water monitoring is hydrogen sulfide (H2S) test kits, which measure the bacterial reduction of thiosulfate [28] but generally require at least 24 hours of incubation and can have limited sensitivity and specificity for *E. coli* and thermotolerant coliforms [29, 30]. Greater specificity can be achieved by using enzyme activity detection [6, 31, 32], surface marker binding [7, 11], or reporter bacteriophages [5, 9]; with enzyme-based tests being the most common due to the possibility of robust signal amplification through one of several available colorimetric, fluorogenic, or electroactive substrates available for $\beta$-galactosidase and $\beta$-glucuronidase, which are both present in *E. coli*. However, all widely-used tests are still limited by a required incubation time of at least 16 hours [15]. Emerging methods decrease the time to results [6, 9, 32], but there is not yet a commercially-available field-based test for rapid detection of *E. coli* in the concentration range of interest for drinking water monitoring.

In order to guide the development of point-of-care medical diagnostics tests for use in resource-limited settings, the WHO Sexually Transmitted Diseases Diagnostics Initiative formulated a set of criteria stating that the ideal such test would be ASSURED (Affordable, Sensitive, Specific, User-friendly, Rapid/Robust, Equipment-free, and Deliverable to end users) [33–36]. The ASSURED criteria constitute requirements that are generally applicable across various point-of-use sensing applications in resource-limited settings. However, they do not capture context-specific requirements, nor do they prescribe a systematic approach to

**Table 1. Translation of sensing technology options to corresponding design attributes as perceived by users, along with the relative cost of each option.**

| Sensing technology | | User design attribute | | Cost |
|---|---|---|---|---|
| Parameter | Option | Parameter | Option | |
| Sensing modality | Basic colorimetric | Sensor output | Visual | Low |
| | Electrochemical | | Electronic | High |
| Signal augmentation | Incubation only | Time to results | >16 h | Low |
| | Concentration | | <4 h | High |
| System architecture | Disposable | Reusability | Low | Low |
| | Mostly reusable | | High | High |
| Reagent introduction | Liquid reagents | Preparation complexity | High | Low |
| | Incorporated reagents | | Low | High |

determine those requirements, necessitating the implementation of a separate methodology to determine design requirements for a specific context and sensing application [36].

Sensing applications may have specific user requirements that are non-obvious to researchers, and these user requirements may affect the sensing technologies required for effective sensing. In the case of point-of-use bacterial sensing in drinking water, we can translate sensing technologies into sensor attributes as perceived by users (Table 1). For example, basic colorimetric detection is relatively inexpensive for simple presence-absence testing due to the possibility of visible inspection [4]; however, these visual results are difficult to interpret quantitatively [37], frequently necessitating the use of techniques such as image processing for accurate quantification [6, 10]. Alternatively, electrochemical detection requires additional cost and complexity but offers straightforward transduction of cell concentration to an electronic signal [38]. Currently, all tests used in the field are incubation-based with a time to results of at least 16 hours [15]; however, there is an increasing interest in using physical concentration to decrease the time to results [5, 6, 9, 39]. System architecture can incorporate a wide range of reusability: inexpensive disposable tests such as existing H2S tests [28] are generally available for less than 1 USD, whereas reusable systems for online monitoring [32, 40] may retail for more than 40,000 USD. In addition, mostly-reusable sensors have been demonstrated [3, 41] which allow for the use of a more expensive electronic component, similar to existing blood glucose meters, combined with an inexpensive disposable component. This disposable component may contain consumable reagents or surfaces to come in contact with biological samples, which may be prohibitively difficult to clean in resource-limited settings. Finally, the complexity of reagent introduction can vary from required addition of liquid reagents to fully incorporated reagents [8].

Despite the importance of considering user needs to identify appropriate sensing technologies, new sensors intended for use in resource-limited environments are often developed with limited interaction with their intended user base, and such interaction generally occurs in the later stages of development (e.g., prototype testing). Field trials at the end of development provide valuable information on real-world sensor performance [6, 12, 42], but the desired performance criteria as identified at the beginning of development are generally filtered through multiple degrees of separation due to the remoteness of the intended users [33, 42]. Moreover, it is valuable to understand the context of how the information generated by sensors will be handled and transduced to decisions at the household, community, and government level [43]. As sensor placement becomes more ubiquitous and integrated with improved informatics systems for data-based decision making via concepts such as Big Data and Internet of Things, it becomes even more crucial to ensure that sensors are developed based on the

requirements of their intended use cases. In the context of environmental monitoring, development of complete environmental measurement solutions requires consideration of the environmental science principles that determine contamination and measurement mechanisms, the sensor attributes that determine usability, and how the resulting information is handled.

Interaction with users and other stakeholders before sensor development is important in order to identify the most promising use cases for new sensors [43] and to set design priorities for a given use case. For example, interaction with clinicians during development of medical technology intended for use in developed nations is common and provides valuable information which guides the direction of research [44], and there is increasing interest in beginning stakeholder engagement early in the development of diagnostics intended for use in resource-limited settings [35, 42]. By taking part in the process of determining user needs, sensor researchers can leverage their unique position of knowing what technological improvements are feasible and inform their design priorities through direct interaction with end users and other key stakeholders, who jointly are most knowledgeable of the requirements for improved sensing solutions. Many sensor researchers would value the opportunity to gain stakeholder feedback at the beginning of development but lack experience in needs assessment and market research [42], necessitating presentation of systematic methodologies for determining design priorities via stakeholder engagement.

In this work, we present a fieldwork-based investigation process to determine design priorities for new sensors. We focus on understanding user needs for bacteriological drinking water tests in rural India, due to the widespread prevalence of microbial water contamination in this context as well as the presence of a network of stakeholders dedicated to remediating this contamination through both technological solutions and behavior change interventions. Stakeholder interviews and meetings are employed to understand how information flows in the context of interest and to identify potential use cases for new sensing technologies, whereas group design workshops are used for an initial identification of design preferences. Conjoint analysis, which is well-established in the assessment of user preferences for applications such as consumer products [45], healthcare decisions [46], and environmental policies [47, 48], is used here to provide a quantitative assessment of the relative importance of different attributes. This work is intended to present our fieldwork-based determination of design priorities for point-of-use bacterial water detection, as well as best practices for other researchers developing sensors for use in resource-limited areas.

## Materials and methods

### Fieldwork overview

The fieldwork workflow involved a multi-step exchange between sensor developers and local stakeholders (Fig 1), beginning with a literature review and hypothesis generation for potential analytes and users for improved point-of-use sensing. We investigated these hypotheses via *Stakeholder Meetings and Interviews* to obtain information on Knowledge, Attitudes, and Practices (KAP) regarding water quality management. Furthermore, *Group Design Workshops* were conducted to identify users' priorities for point-of-use sensors. Together, these activities produced an early sense of use cases which could be assessed for technical feasibility. We returned to the field with this understanding of the feasible options and conducted *Conjoint Analysis Interviews* to quantify user priorities for improved point-of-use sensors.

Fieldwork-based research occurred over three visits to India: visits to Maharashtra and Jharkhand states in January 2016, a visit to Uttarakhand state in August 2016, and visits to Uttarakhand and Jharkhand in January 2017 (study site locations in Table A of S1 Appendix). All human-subjects research procedures were approved by he Massachusetts Institute of

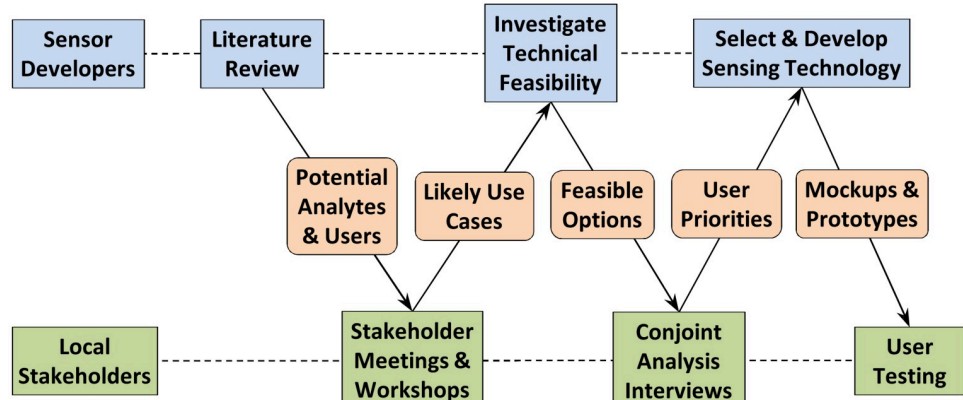

**Fig 1. Proposed sensor development workflow incorporating fieldwork-based determination of design priorities.**

Technology Committee on the Use of Human Subjects (MIT COUHES) under approved protocols 1511312247 and 1511312247A001. Translated consent forms were used for fully literate interview participants. For participants with limited literacy, verbal consent was used in place of consent forms in order to avoid unnecessary distress due to lack of understanding of the consent form contents. In cases of verbal consent, participants provided consent after being explicitly told the intent of the interview or workshop, that their participation was completely voluntary, and that they were free to end their participation at any time. Our decision to seek human-subjects research approval via MIT COUHES ensured that we offered the same level of protection to human subjects in India as that offered to human subjects involved in any federally funded research in the United States and is consistent with Indian Council of Medical Research (ICMR) guidelines that social and behavioral research for health applications should be approved by the ethics committee for the researchers' institution [49].

## Stakeholder meetings and interviews

Stakeholder meetings included meetings with members of both governmental agencies and non-governmental organizations (NGOs), and allowed for institutional and policy mapping [50] of water quality management [51]. We met with state officials from agencies associated with water quality management; local government officials charged with water quality management, public health, and village governance; and NGOs working with rural communities on water quality management, with a full description of stakeholder meetings provided in S1 Appendix.

Knowledge, Attitude, Practice (KAP) interviews [52, 53] were conducted with five individuals in Shedashi, Maharashtra, as well as a group interview with eight PRADAN staff members in Torpa, Jharkhand. Questions included demographic information, knowledge of water sources and water quality management, attitudes regarding water quality, and current water quality practices, with interview text and anonymized responses provided in S1 File. In order to gauge comfort and attitudes regarding point-of-use water testing, participants were also shown a demonstration of point-of-use measurement of total dissolved solids (TDS) in a water sample using a commercial TDS meter (AP-1 AquaPro, HM Digital, Redondo Beach, California, USA), given the opportunity to test the water themselves, and asked their attitudes regarding the ease of using the test, how they would use the information from a test like this, and how often they would use such a test.

## Group design workshops

We conducted five group design workshops for an initial identification of design priorities with end users and NGO staff. The size of each workshop varied from 8 to 37 participants, with a total of 71 participants (workshop locations and summary statistics in Tables A and B, respectively, of S1 Appendix). All participants were recruited through our NGO partners (RC-CEL, PRADAN, ACE). During these workshops, participants were presented with hierarchy cards (Fig 2) corresponding to options for key design attributes: level of ownership ("Owned at household level" or "Owned at community level"), time to results ("Same-day results"), sensor output ("Tells simple presence or absence", "Tells amount of contaminant", or "Tells amount of contaminant and recommended action"), reusability ("Reusable" or "Disposable"), and complexity of use ("No mixing required"). Level of ownership was selected as a proxy for cost, due to concerns that workshop participants may not reveal accurate pricing in a workshop situation due to discomfort in revealing their financial status in a group context or concern that they were negotiating a purchase of a yet-to-be-created sensor. In addition, level of ownership preference allowed for evaluation of preferred use cases, as participants would indicate whether they were interested in individual sensor ownership.

These attributes were selected due to their potential for translation to appropriate sensing technologies for use in improved point-of-use sensing (Table 1), as discussed further in the Results and discussion section. Five additional hierarchy cards ("immediate results", "no control or calibration solution required", "number output", "color change", and "light output") were used for the first two group design workshops, but were then eliminated due to redundancy ("immediate results" with "same-day results"), difficulty in explaining the concept to the participants ("no control or calibration solution required"), and low perceived priority among participants ("number output", "color change", and "light output").

Participants selected the hierarchy cards corresponding to what they considered to be the three most important attributes, with the selected design priorities tabulated across all workshops to yield overall design priorities. Participants were asked to vote for their preference for each design area, then to select the three hierarchy cards corresponding to what they considered to be the three most important attributes. The selection of the three most important attributes was done either individually or collectively as a group, with the priority selection procedure varied depending on the group dynamics of each workshop and if it was feasible to acquire independent information on each participant's design priorities. For individual selection, each participant selected three non-ordered hierarchy cards corresponding to what they considered to be the most important attributes, with overall design priorities for the workshop defined as the attributes selected by the most participants. For group selection, the participants collectively reached a consensus on which attributes were the first-, second-, and third-most important. In order to determine the overall design priorities across all workshops, the top three priorities from each workshop were given points, with 5 points for the first-most important, 4 points for the second-most important, and 3 points for the third-most important attribute. The points for each attribute were then summed over all five workshops (Table G of S1 Appendix) to determine the overall ranking of design priorities (Fig 2).

During the workshops, participants were also shown a demonstration of two existing point-of-use water tests, the reusable TDS meter used for the KAP interviews and disposable pH strips (Universal Indicator Paper, Tzakzy, Taizhou, PRC); given the opportunity to test the water themselves; and asked what they perceived to be the tests' advantages and disadvantages. Participants were also asked introductory and concluding questions regarding their knowledge, attitudes, and practices regarding water quality management, with a sample workshop runsheet included in S3 File.

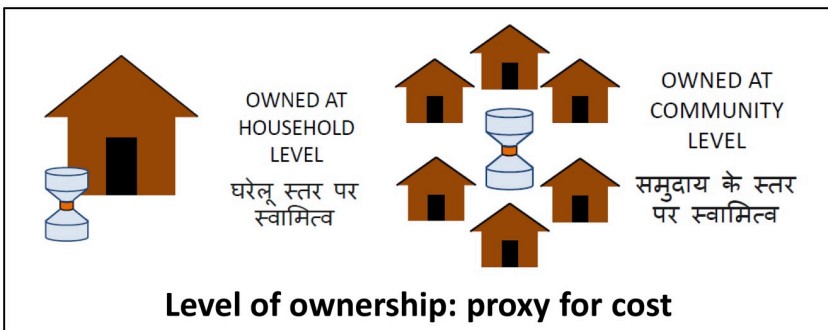

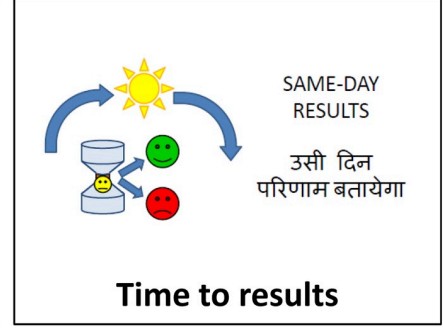

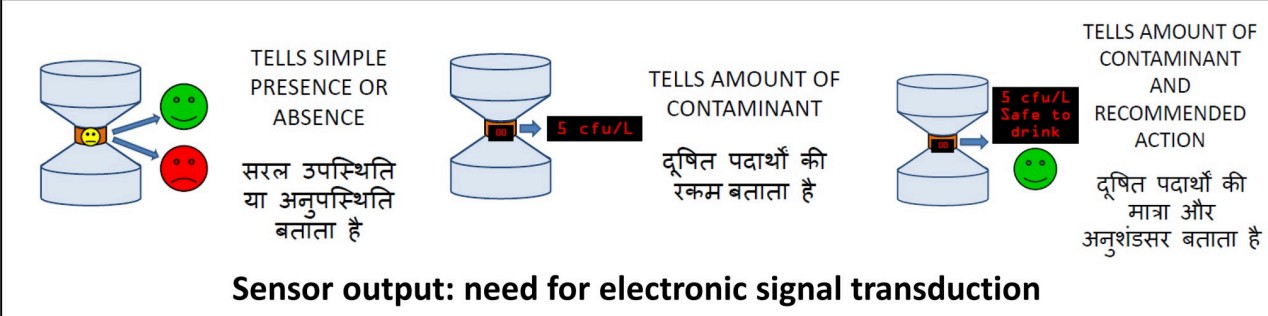

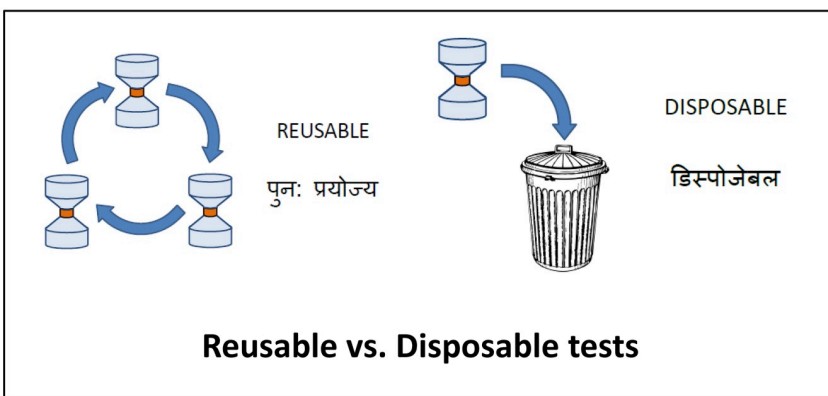

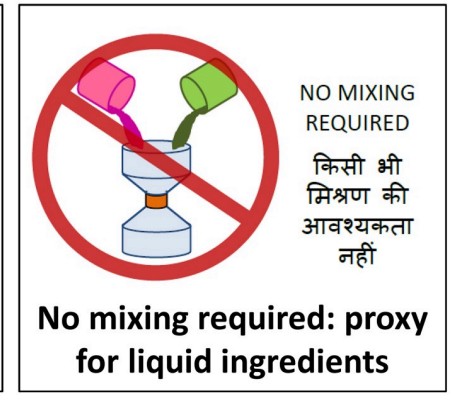

## Overall design priorities

1.  Owned at household level

2.  Tells amount of contaminant and recommended action

3.  Same-day or immediate results

4.  Reusable tests

**Fig 2. Final set of hierarchy cards used for initial identification of design priorities in group design workshops, as well as the attributes identified as overall design priorities from the group design workshops.**

### Conjoint analysis study design

We used choice-based conjoint analysis to quantitatively validate the design priorities identified in the group design workshops and evaluate their relative importance. Conjoint analysis is

intended to model users' decisions between options with different combinations of attributes [48, 54–56], and has been used to evaluate consumer products for water treatment [45] and environmental policy options [47], among other applications. Conjoint analysis models user choice by treating each available option as having a total utility equal to the sum of the marginal utility for each of its attributes, with the probability of choosing an option determined by the total utility.

Conjoint analysis is intended to model users' decisions among a finite set of options with different combinations of attributes, and has been cross-validated over several studies as a predictive tool of user choice [55]. Each option $i$ is treated as having a utility $U_i$, which is a linear combination of the marginal utilities $u_j$ for each of the $N$ possible attributes [48, 54, 56]:

$$U_i = \sum_{j}^{N} a_{ij} u_j \tag{1}$$

In this work all attributes are modeled as binary entities, so study design coefficient $a_{ij}$ is equal to 1 if attribute $j$ is present in option $i$ or 0 if it is absent. Possible attributes for an option are generally grouped into different options for one overall design attribute (such as cost); these options are referred to as levels of the attribute. The marginal utilities for each attribute level are generally calculated to sum to zero, so that $u_1 = -u_2$ for an attribute with two levels and two corresponding marginal utilities $u_1$ and $u_2$. The probability $P_i$ of a user selecting a given option $i$ from among $M$ availably options is equal to the exponential of its total utility $U_i$ normalized by the sum of the exponential of the total utility for all $M$ options [56]:

$$P_i = \frac{exp(U_i)}{\sum_{i}^{M} exp(U_i)} \tag{2}$$

We investigated a total of five design priorities (Table 2): the four priorities identified from the group design workshops, as well as ingredient addition, identified as a concern during the workshop qualitative responses. For reusability, we included levels for "Disposable" and "Mostly Reusable" tests, due to the considerable complexity required for integrating cleaning procedures into microbial sensing systems for a fully reusable sensor. For sensor output, we

**Table 2. Sensor attributes and levels for pilot and full conjoint analysis interviews.** Sensor attributes and levels are listed along with the prior mean and variance of Level 1 for each attribute used for generation of the study design as described in the methods section. For pilot interviews, costs for mostly reusable tests are the cost for the reusable component followed by (indicated by '/') the cost for the disposable component. For full interviews, cost per test is the average cost per test over 20 tests, with the mostly reusable test costs also listed as the costs of the reusable and disposable components.

| Interview | Attribute | Level 1 | Level 2 | Level 3 | Prior mean | Prior variance |
|---|---|---|---|---|---|---|
| Pilot | Reusability | Disposable | Mostly Reusable | | 0.00 | 1.00 |
| | Output | Amount | Amount + Recommendation | | -1.00 | 1.00 |
| | Time to results | Next Day | Same Day | | -1.00 | 1.00 |
| | Ingredient addition | Add Liquid | Add Tablet | All Ingredients Combined | -1.00 | 1.00 |
| | Disposable test cost, ₹ | 100 | 50 | | -1.00 | 1.00 |
| | Mostly reusable test costs, ₹ | 1000/50 | 500/10 | | -1.00 | 1.00 |
| Full | Reusability | Disposable | Mostly Reusable | | -0.28 | 0.022 |
| | Output | Amount | Amount + Recommendation | | -0.27 | 0.027 |
| | Time to results | Next Day | Same Day | | -0.26 | 0.026 |
| | Ingredient addition | Add Liquid | All Ingredients Combined | | -0.44 | 0.065 |
| | Cost per test, ₹ (Mostly reusable test costs, ₹) | 50 (500/25) | 100 (1000/50) | | 0.33 | 0.027 |

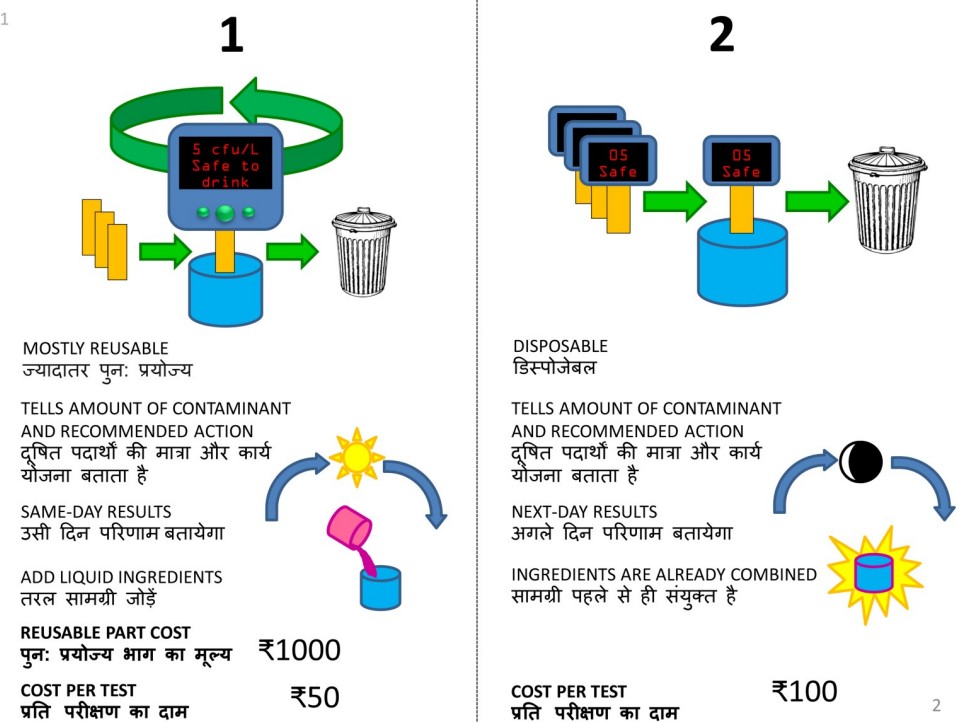

**Fig 3. Sample card used for choice-based conjoint analysis interviews.** Each conjoint analysis card consists of a choice set containing two profiles for hypothetical bacterial water tests. During each conjoint analysis interview, the participant sequentially considered eight such choice sets and indicated their prefered hypothetical bacterial water test between the two options.

included levels for "Amount" and "Amount + Recommendation" to gauge preference for integrated recommendations. For time to results, we included levels for "Next Day" and "Same Day". For ingredient addition, we included levels for "Add Liquid" and "All Ingredients Combined", as well as an "Add Tablet" level which was included in the pilot interviews but removed from the full interviews in order to increase the interviews' statistical power for a given sample size, based on the pilot interview results and our sample size calculations as discussed below. Costs levels (in Indian Rupees, ₹) were selected based on the cost of existing comparable sensors and suggested pricing from stakeholders, with a full discussion of cost level selection provided in S1 Appendix.

For both the pilot and full conjoint analysis interviews, we used JMP (Version 12, SAS Institute, Cary, North Carolina, USA) to generate four surveys consisting of eight choice sets, with choice sets for the full conjoint analysis interviews listed in Table E of S1 Appendix. Each choice set consisted of a decision between two hypothetical bacterial water tests, with three of the five attributes varied between the two options (sample card shown in Fig 3). The choice sets were generated using the Bayesian D-optimality criterion [56] as implemented in JMP, with prior means and variances as specified in Table 2 of the main text. Prior means for the pilot interview choice sets were specified as 0.00 for reusability, corresponding to no expected preference between disposable and reusable tests, and -1.00 for Level 1 of all other attributes, corresponding to a probability of $e^{-1}/(e^{-1} + e^1) \approx 12\%$ for choosing the test with Level 1 of that attribute over one with Level 2 of that attribute if all other attributes were the same between the two options.

We used the observed marginal utilities from our pilot conjoint analysis to generate the choice sets for our full conjoint analysis interview, as well as to calculate the required sample size for the full conjoint analysis interviews. The prior mean and variance of Level 1 for each attribute in the full interview choice sets was specified as the parameter estimate and variance of the corresponding attribute level from the pilot interviews. The means and variances from the pilot interview results were also used to calculate the required sample size for the full conjoint analysis interviews according to the method described by de Bekker-Grob et al. [46] using a two-tailed false-positive probability $\alpha = 0.05$ and statistical powers of 0.8 (i.e., false-negative probability $\beta = 0.2$) and 0.9 ($\beta = 0.1$), with sample size calculations shown in Table F of S1 Appendix. The conjoint analysis responses were analyzed in JMP (Version 12, confirmed in Version 13) using conditional logistic regression to obtain Firth bias-adjusted maximum likelihood estimates (MLEs) of the marginal utilities for all attribute levels.

### Conjoint analysis interviews

Conjoint analysis interviews were conducted in two rounds, with pilot interviews (N = 10) in Uttarakhand in August 2016 and full interviews (N = 53) in Uttarakhand (N = 28) and Jharkhand (N = 25) in January 2017 (interview locations in Table A of S1 Appendix, summary statistics in Tables C and D of S1 Appendix). The full interview consisted of acquiring demographic information; introductory and concluding questions regarding knowledge, attitudes, and practices about water quality management; and one of the four conjoint analysis surveys, with the four surveys cycled through for consecutive interviews. For the conjoint analysis survey portion of the interview, the participant was sequentially shown the eight cards corresponding to the choice sets for that survey (sample card shown in Fig 3, all cards for full conjoint analysis interviews provided in S5 File). For each card, the participant indicated which hypothetical bacterial water quality test they would prefer between the two available options. In order to effectively communicate different sensor architectures, participants were shown pH strips as an example of a disposable test, a blood glucose meter (CVS/pharmacy Advanced Glucose Meter, AgaMatrix, Salem, NH, USA) as an example of a mostly reusable test, and a bottle of Phenol Red pH test solution (JED Pool Tools, Scranton, PA, USA) as an example of a test requiring the addition of liquid. Interviews were generally completed in 30-40 minutes, ensuring that participants had sufficient attention to evaluate all choice sets in the conjoint analysis survey.

Pilot interview participants consisted of NGO staff, all interviewed at the Himmotthan office in Dehradun, and end users recruited through our NGO partner (Himmotthan). Full interview participants consisted of the main population of end users (N = 45) and a separate population of local contacts (N = 8). End user participants were recruited through our NGO partners (Himmotthan and PRADAN), either door-to-door or through group model-building workshops simultaneously conducted to evaluate system dynamics of water quality management [57]. End user participants were preferentially selected to be women (73% of total participants) due to heavily gendered responsibility for household water management, and particularly from women enrolled in women's self-help groups (SHGs, 64% of total participants), due to their key role in implementing new water projects and practices in rural communities.

The conjoint analysis responses were analyzed in JMP to calculate marginal utilities of all attribute levels. We also conducted a preliminary investigation of market segmentation by evaluating demographic interactions with design attributes for the main population of full conjoint analysis interview participants (S1 Appendix). Statistical significance for all estimates was evaluated using Likelihood Ratio tests, with estimates with $p < 0.05$ denoted as significant and

estimates with $p < 0.005$ denoted as highly significant for consistency with recommendations for reporting the significance of new effects [58].

## Results and discussion

We structured our investigation based on initial assumptions of water quality management needs that we identified from peer-reviewed literature. Through fieldwork-based research, we aimed to develop an improved, context-dependent understanding of likely use cases and user priorities of the sensors that would be suitable for guiding researchers in the selection of appropriate sensing technologies. At the onset of our initial study design, we assumed that microbial contamination would be the most suitable contamination for improved point-of-use monitoring due to the extended time to results of existing microbial water tests [15], as well as the widespread nature of microbial water contamination in India and globally [16, 17, 19, 59, 60]. Conversely, chemical contamination is generally localized due to the heterogeneity of its geogenic and anthropogenic causes [60–64]. Furthermore, we assumed that the most likely use cases for improved point-of-use testing would be as part of government-mandated water quality testing, either for routine testing [65–68] or for additional testing such as the investigation of outbreaks [69] or exposure pathways [70], as the government remains the largest custodian of water as a public good in India [71].

### Stakeholder meetings and interviews

Our stakeholder meetings served to provide context of water quality management practices and current policies, resulting in an understanding of likely users and use cases. From these stakeholder meetings, we learned that Indian water quality management is predominantly driven at the community level, creating a promising potential for improved point-of-use testing. Planning and implementation of water and sanitation projects occurs primarily at the village (Gram Panchayat) level, consistent with the Panchayat Raj model of governance. Funding for these projects is available through grants and reimbursements from higher levels of government, such as the Swachh Bharat program for sanitation projects (which is administered at the national level) or the National Rural Drinking Water Programme (which is administered at the state level). For projects related to water, local communities are responsible for initiating projects and requesting funding. These local communities are assisted and informed by a network of NGOs who conduct awareness campaigns related to water and sanitation and provide technical expertise for project planning. Community water planning has traditionally occurred through the local village councils (Gram Panchayats), but in places where local women's self-help groups (SHGs) are sufficiently organized, they often advocate for improving water systems, and in some circumstances will initiate water and sanitation projects and apply directly to government agencies for project funding as part of their role of organizing and advocating for community needs.

Microbial water contamination is widespread in India, but we learned that existing point-of-use bacterial water tests are generally insufficient for either awareness campaigns or community advocacy. We spoke to NGO workers (Himmotthan) who had tried to use existing point-of-use tests for awareness campaigns, but found them unsuitable due to the need for 16-48 hours of incubation before results are available, making them difficult to incorporate into behavior change interventions. Furthermore, we learned from meeting with NGO staff (Arghyam, WaterAid) that existing H2S tests do not have sufficient sensitivity or specificity to mandate government action without additional lab-based testing [65, 66], making them unsuitable for government advocacy. Development of improved point-of-use sensors with decreased time to results and satisfactory sensitivity and selectivity would enable incorporation

into behavior change interventions, facilitating the development of new evidence-based behavior change intervention methods. Based on this strong need for improved point-of-use testing among NGOs, we decided to focus on NGO-led behavior change interventions as our initial primary use case, with the potential to spread to additional use cases among NGO staff and the rural residents that they interact with (Fig A of S1 Appendix).

From our individual KAP interviews, we learned that many rural residents are aware that poor water quality results in disease, but are unaware of water quality management practices aside from sensory evaluation (via taste, smell, and visual appearance) and routine treatment (generally via boiling or chlorination). Moreover, rural residents indicated a strong interest in both knowing when their water is contaminated and what remediative actions to take when it is contaminated. During demonstrations of existing TDS meters, participants immediately requested recommendations of what to do based on the measured TDS of their water, highlighting a strong interest in both point-of-use testing and integrated reporting of contaminants and recommended actions. Interview participants also indicated a preference for reusable sensors over disposable tests, with a local government official in Karanjtoli, Jharkhand stating that residents could be willing to pay as much as ₹1000 for a reusable sensor.

## Group design workshops

Our group design workshops provided an initial identification of user preferences for further investigation. We considered key sensor attributes that would be apparent to users during device operation and also require specific technical decisions early in the sensor design process (Table 1). For example, dimensions and form factor were not considered, as these can be achieved via multiple sensing modalities and are considered later in development during the product design phase. Before beginning our design workshops, we hypothesized that users would prefer low-cost, disposable, presence-absence tests offering same-day results, i.e. the same attributes as existing point-of-use tests with a decrease in time to results. Our hypothesized user preference for a simple presence-absence test was based on both the output of existing point-of-use tests, which generally only report microbial contamination as present or absent, and a stakeholder interview with a government official who believed that end users would not be interested in more sophisticated sensor outputs.

Participants' priorities from all design workshops were combined as described in the methods section to yield an initial identification of sensor design priorities (Fig 2): 1) owned at household level, 2) tells amount of contaminant and recommended action, 3) same-day or immediate results, and 4) reusable tests. In addition, qualitative responses indicated significant concerns about liquid reagents related to their potential toxicity, complexity associated with their handling, and availability of consumable liquid reagents in remote areas. The interest in household ownership was consistent with our hypothesized interest in low-cost sensors, and motivated consideration of use cases involving individual testing, as well as technology selection aimed at developing sensors simple enough for rural end users to operate and interpret. The interest in same-day or immediate results was consistent with our hypothesized preference for fast results. However, the preference for output of amount and recommended action was in stark contrast to our hypothesized user preference for a simple presence-absence test. Moreover, the preference for a reusable test with a higher initial cost but lower cost per test over time was unexpected and suggested a need for development of mostly reusable sensor architectures for bacterial sensing.

Based on the design workshop results, sensor cost, sensor output, time to results, reusability, and requirement for liquid reagents were selected as attributes to investigate via conjoint analysis interviews.

## Conjoint analysis interviews

Conjoint analysis interviews allowed us to quantitatively validate and prioritize the preferences observed in our group design workshops. We hypothesized that the conjoint analysis interviews would reveal the same preferences as our design workshops: low cost, output of both amount of contaminant and recommended action, same-day results, mostly reusable tests, and combined ingredients with no need for addition of liquid reagents.

The results of our conjoint analysis interviews (Fig 4) indeed showed preferences consistent with the priorities from the group design workshops. The pilot conjoint analysis interviews (Fig 4a) showed agreement with the group design workshops, even for a small population in a different state than where the workshops were conducted, and provided initial utility estimates that we used to design the full conjoint analysis study. The full conjoint analysis interviews of our main population of end users (Fig 4b) continued this agreement, with highly significant preferences ($p < 0.005$) for mostly reusable tests, integrated output of contaminant amount and recommended action, same-day results, and low cost, as well as a significant preference ($p < 0.05$) for combined ingredients with no need for liquid reagent addition. The highest measured utility was for output of amount and recommended action, highlighting the importance of integrating test results and recommendations as initially observed during the individual KAP interviews.

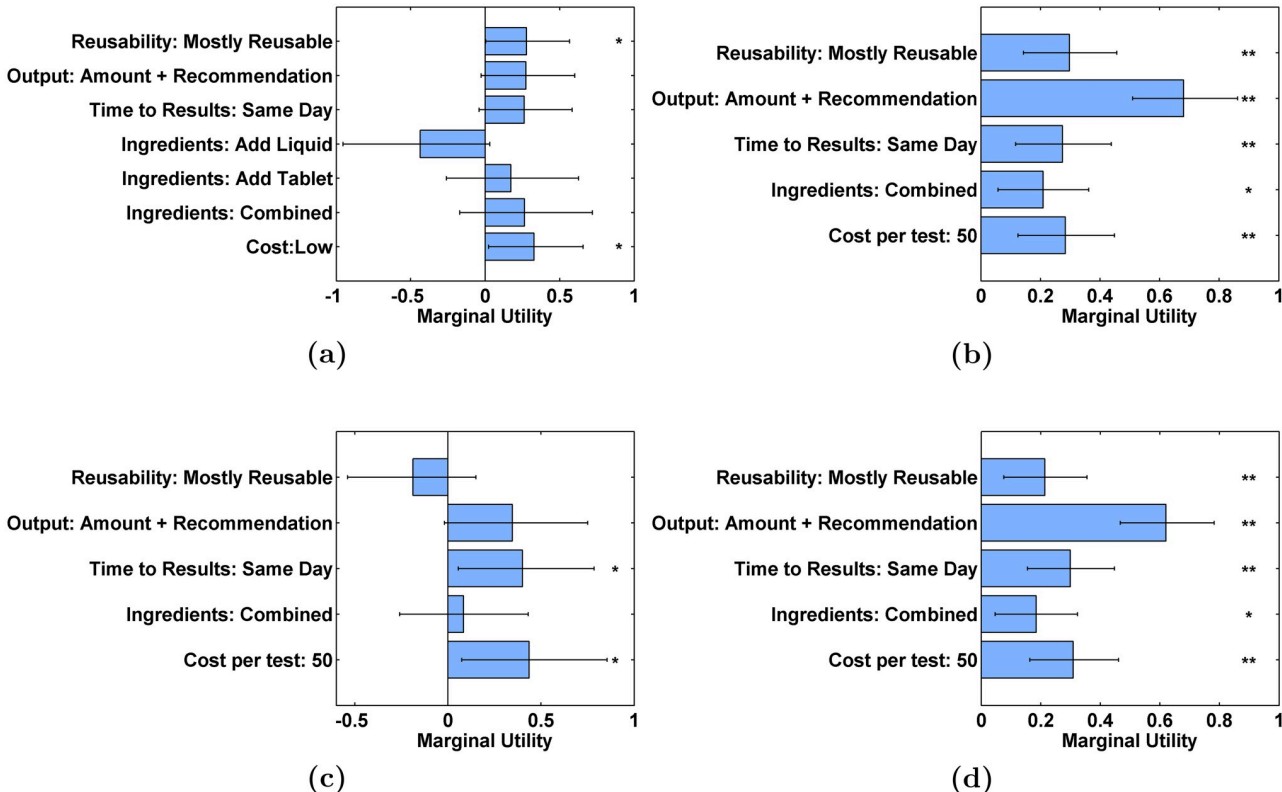

**Fig 4. Conjoint analysis results.** Conjoint analysis parameter estimates for (a) pilot conjoint analysis interviews (N = 10) (b) full conjoint analysis interviews of main population (no local contacts, N = 45) (c) full conjoint analysis interviews of local contacts (N = 8) (d) full conjoint analysis interviews of entire population (main population and local contacts, N = 53). Error bars are 95% confidence intervals. Asterisks are significance level, where * denotes $p < 0.05$ and ** denotes $p < 0.005$ according to a likelihood ratio test on each parameter estimate. Parameter estimates are tabulated for all levels in Tables H, I, J and K of S1 Appendix.

The full conjoint analysis interviews for local contacts (Fig 4c) were considered separately due to the local contacts' higher education level (all local contacts had at least 12 years of education, whereas only 24% of the main population had at least 12 years of education) and decreased isolation relative to end users. We observed significant preferences for same-day results and low cost, similar to end users, but did not observe significant preferences for mostly reusable tests, integrated recommendations, or combined ingredients. The lower significance is expected for a smaller population size, and the decreased utility estimates are consistent with a higher level of awareness about remediation techniques and decreased concerns about the logistics of procuring supplies and reagents. When the full conjoint analysis interviews are considered for the entire population, consisting of both the main population and local contacts, the resulting utility estimates (Fig 4d) are largely the same as for the main population considered alone, with a slight decrease (approximately equal to the standard error of the estimate) in the estimated utility of mostly reusable tests, with all other estimated marginal utilities well within one standard error of the estimate between the two populations.

## Identification of appropriate sensing technologies from user preferences

We can use our findings to identify which bacterial sensing technologies are most appropriate for improved point-of-use bacteriological water sensors. The observed strong interest in integrated reporting of contamination and recommendations necessitates electronic transduction of results, with the resulting electronic signal converted to recommendations which could be displayed on a sensor readout in parallel with contamination level as shown in the hypothetical water tests on the sample conjoint analysis card (Fig 3). These recommendations would be informed by context-dependent knowledge of the best available remediation actions for a given contaminant level, and could even be programmable for a given sensor so that local sensor distributors could best integrate context-dependent knowledge.

Electronic transduction of results can be accomplished via electrochemical detection [38], an approach that can leverage the extensive development of low-cost electrochemical blood glucose meters [41]. Colorimetric detection generally requires the use of cloud-based image-processing capabilities in order to achieve electrical transduction of bacterial contamination at the WHO guideline of 0 cfu/100 mL [6, 10], but advances in low-cost light-emitting diode (LED)-based spectrophotometers [72–74] and spectrophotometers utilizing smartphone optics [75, 76] may lead to low-cost colorimetric sensors offering electrical transduction at the required sensitivity without the need for image processing.

As shown in the proposed sensor development workflow (Fig 1), additional refinement of the selected sensing modality can take place through user testing of mockups and functional prototypes that are developed based on results of stakeholder meetings, design workshops, and conjoint analysis interviews. User testing of prototypes or mockups can incorporate both usability studies as well as evaluation of the technical performance of functional prototypes under their intended use conditions [30, 77, 78]. In this way, researchers can evaluate user attitudes towards factors such as controls and calibrations that are difficult to explain or assess in the absence of a functional prototype.

We also observed a strong interest in same-day results for awareness campaigns and community advocacy. Same-day detection will likely require physical concentration of bacteria in water samples [5, 6, 8, 9], which will increase the cost and complexity of the sensor. In use cases where a detection time of a few hours is acceptable, improved bacteriological tests could use a combination of physical concentration and incubation to increase sensitivity. Observed openness to mostly reusable tests allows for the use of reusable electronic components for electrochemical sensors [41] or low-cost spectrophotometers [74]. Moreover, interest in combined

ingredients necessitates development of tests with reagents incorporated into the sensing region [8]. These conclusions from our fieldwork-based stakeholder engagement motivate the development of low-cost point-of-use bacteriological water tests which are suited to likely use cases in rural India, such as evidence-based behavior change interventions or needs assessment by local NGOs, community-led advocacy by local stakeholders such as women's self-help groups, as well as operation and maintenance of completed water treatment systems in resource-limited settings (Fig A of S1 Appendix).

## Best practices for fieldwork determination of sensor design priorities

Our fieldwork-based research also helped identify best practices for sensor researchers interested in fieldwork-based determination of design priorities for other environmental sensing needs. We found that it was helpful to acquire information 'in stereo' by independently speaking to different stakeholders within same community. For example, in Shedashi, Maharashtra, we met with local government officials through meetings with the village council (Gram Panchayat) and local Public Health Center (PHC); conducted interviews and a design workshop with end users; and conducted a design workshop with NGO staff who work with the community. This stereo investigation provides valuable information in multiple useful ways: about how different stakeholders perceive the same situation, where they agree or disagree, and how information flows in a given social structure. This understanding enables effective selection of sensor use cases, especially where to inject information into a social structure.

We also found that we could learn invaluable information about how residents would respond to a new point-of-use sensor by demonstrating existing sensors that output information similar to the intended output of the sensor to be developed. We observed that rural residents presented with access to a TDS meter would go back to their own water sources to collect samples to be tested, emphasizing a strong desire to see directly if their water was contaminated. However, demonstrating a test which provides actual results raises challenges about what researchers should do when they measure contamination, as residents whose water had higher TDS levels immediately wanted to know if their water was safe to drink and what remediation was necessary.

Similarly, we found that it was valuable to bring physical prototypes to stakeholder meetings, even if the sensor architecture had not yet been finalized. In our meetings, a commercial disposable filter unit (Nalgene Rapid-Flow, 250 mL volume, Nalge Nunc International, Rochester, NY, USA), selected because of a general similarity in appearance to potential filtration-based sensor architectures, brought out valuable questions and concerns about how such a device would be used. In hierarchical settings such as government agency meetings, prototype use spurred discussion from a wider range of participants, as technicians and support staff indicated their respective needs for an improved sensor.

A team that seeks to conduct fieldwork-based research will likely benefit from expertise in both sensor development and social science research methodology, as can be accomplished by incorporating team members from fields such as city and regional planning, policy studies, and public health. Combining expertise in both sensor development and social science research into one team allows for effective translation of sensor attributes to sensing technologies and vice versa, while simultaneously identifying use cases grounded in an understanding of how information and decision-making are handled in a given context.

## Recommendations for identifying and interacting with local partner organizations

In planning and implementing fieldwork-based needs assessment, it is crucial to partner with local organizations doing development work in the environment of interest. Such partnerships

allows the academic researcher to utilize the partner's relationship with the communities in arranging the interviews, workshops, etc. Moreover, organization staff provide their own valuable perspective on the needs and challenges of the communities that they work with, and can help provide a path to implementation of a completed measurement solution.

One of the key challenges for laboratory-based researchers developing sensors for use in resource-limited settings is building connections with local partner organizations in order to identify sensor use cases and design priorities [42]. At academic institutions, one of the most promising ways to connect with local partner organizations is to reach out to researchers from disciplines with established connections to stakeholders working in contexts similar to the context of interest for sensor development, as is common for researchers working in fields such as regional planning or public health. Even if these existing stakeholder connections are not working on the exact problem of interest, they are likely to be able to provide connections to stakeholders more closely associated with the application for which the sensor is to be developed, and if they are working in the application space of interest then they may be able to facilitate connections with other stakeholders who can provide a different perspective. The initial availability of connections may vary widely for researchers depending on their institution, location, and area of expertise, but reaching out to existing professional contacts remains one of the most promising ways to connect with local stakeholders and motivates the development of collaborations and conferences intended to connect sensor researchers with stakeholders seeking sensing solutions for challenges in resource-limited contexts.

In our study, our entry point was an initial connection with the agency responsible for groundwater management in the state of Maharashtra (Maharashtra GSDA) through colleagues in our institution's architecture department who have worked with this agency on developing decision support tools for water quantity management [79, 80]. This agency provided valuable information on government-mandated testing of groundwater, and also provided connections to the agency responsible for monitoring water for pollutant contamination (Maharashtra Pollution Control Board) which provided an additional valuable perspective on policies and institutions for monitoring of anthropogenic contamination in water.

Similarly, in Jharkhand we connected with our primary NGO partner (PRADAN) through colleagues who had worked with PRADAN on the development of solar-powered irrigation systems as part of PRADAN's agricultural work with rural communities [81]. We in turn were able to benefit from PRADAN's knowledge of water and sanitation management, and PRADAN staff facilitated our connections to our other partners in Jharkhand (ACE and UNICEF), as well as arranging our stakeholder meetings, KAP interview, design workshops, and conjoint analysis interviews in Jharkhand. From this process, we learned that one of the most valuable practices for researchers conducing stakeholder engagement is to ask stakeholders who else it would be beneficial to talk to, and pursue these pathways to identify a network of local partners.

In Uttarakhand, we connected with our primary NGO partner (Himmotthan) as one of the coauthors of this paper worked with them in the development of improved point-of-use soil nutrient management [3, 82]. It should be noted that both PRADAN and Himmotthan are also grantees of the Tata Trusts, who are the main sponsors of our research. While designing a rigorous research protocol allowed us to remain unbiased, having a common connection to the Tata Trusts made it easier for both organizations to allocate human resources for helping us make a connection with the communities they work with.

In engaging with local partner organizations, it is absolutely critical to do so in a mutually beneficial manner. Facilitating stakeholder meetings, interviews, and workshops can require a considerable time commitment for organizations which are already limited by staff and resources, and it is critical to ensure a clear understanding of what researchers and local

partner organizations expect from each other throughout their interaction. In planning and implementing research activities, it is recommended to design research methods that provide benefit to both the sensor developers and local partners while minimizing burden on both local partners and participants in human-subjects research. One of the best ways to do this is to design interviews and workshops such that the resulting information is useful for both sensor developers and local partner organizations, who may benefit greatly from additional information about the awareness, preferences, and practices of the local residents who they work with. In doing this, it may be helpful to design research projects with multiple outputs, including both a sensor that will take time to develop and other outputs such as decision support tools which will assist local partner organizations in their ongoing work [57, 82]. In cases where the immediate benefits to partner organizations will not commensurate with the required cost of personnel time and other researchers, it is important to consider whether it is most appropriate to financially compensate local partner organizations for their assistance.

Furthermore, ideally research activities should be developed that benefit the local participants in human-subjects research. Group workshops can provide a valuable opportunity for both gathering information and spurring discussion among local participants, and interviews and workshops can be developed to convey information to participants after learning the participants' knowledge and preferences.

Finally, in research where the focus traverses between field and laboratory, it is critical to keep in touch with field partners even when the sensor developers are immersed in the lab research. As discussed above, intermediate deliverables such as term papers, research posters, and lab presentations are very useful to and easy to share with partners to keep a sense of continuity. Overall, it is useful if both sides take a longer-term perspective and develop collegiality to exchange competencies useful to each other rather than taking a limited view of a single or a series of field visits. It is this kind of nurturing connection that is vital in connecting the lab to the land and vice versa.

## Conclusions

Our fieldwork-based methodology demonstrates a systematic approach for determining context-specific design priorities for sensing applications intended for use in resource-limited settings, facilitating identification of appropriate sensing technologies at the onset of sensor development. Through our stakeholder meetings and interviews, we identified a need for improved point-of-use bacterial water tests in behavior change communication and other use cases associated with nongovernmental organizations (NGOs) working with rural residents on sanitation and water quality management. Through our design workshops and conjoint analysis interviews, we identified a strong preference among rural end users for integrated reporting of contamination and recommended actions, as well as preferences for same-day bacterial sensing, mostly reusable sensor architectures, combined ingredients, and low cost per test. We then translated these results to identification of appropriate sensor technologies, motivating the development of mostly reusable sensors incorporating physical sample concentration and utilizing sensing modalities, such as electrochemical detection, which are readily transduced to an electronic signal that can be processed to output contamination quantity and recommended remediation actions.

Fieldwork-based determination of sensor design priorities is a cost-effective way to guide the development of appropriate sensors for resource-limited environments. The cost of incorporating a three-week field visit into a research project is significant, at around 2000-3000 USD per researcher including airfare, but this comparable to one month's stipend for a graduate student at many research universities in developed countries. Compared to the cost of

spending years developing a sensor that potentially does not meet users' needs, incorporation of field visits before and during sensor development is a cost-effective strategy for increasing the likelihood of developing an appropriate sensing technology during the first development attempt. Meeting with users in resource-limited environments to determine their needs is achievable and drastically improves chances of developing a sensor that meets users' needs.

## Supporting information

**S1 Appendix. Main supporting information.** Comprises the following sections, containing thirteen tables and two figures:

- Locations of study sites

- Stakeholder meetings description

- Demographic information

- Conjoint analysis study design and sample size calculations

- Conjoint analysis cost level selection

- Potential use cases for improved point-of-use testing

- Tabulated results for design workshops and conjoint analysis interviews

- Conjoint analysis with demographic interaction effects.
  (PDF)

**S1 File. Interview responses for Knowledge, Attitude, Practice (KAP) interviews.** Responses are anonymized and tabulated.
(PDF)

**S2 File. Sample runsheet for Knowledge, Attitude, Practice (KAP) interviews.**
(PDF)

**S3 File. Sample runsheet for design workshops.**
(PDF)

**S4 File. Sample runsheet for full conjoint analysis interviews.**
(PDF)

**S5 File. Complete set of conjoint analysis cards for full conjoint analysis.**
(PDF)

## Acknowledgments

The authors thank Ramchander Chepyala and Krithika Ramchander for assistance in translating the text for the design workshop and conjoint analysis interview cards; Sahil Shah and Ron Rosenberg for helpful guidance regarding designing the conjoint analysis interviews and design workshops; Susanna Kahn and members of the MIT Microfluidics & Nanofluidics Research Laboratory for helpful discussions; and Prof. James Wescoat for arranging connections to stakeholders in India.

The authors thank staff from Professional Assistance For Development Action (PRADAN), Himmotthan Society, Tata Trusts, Rural Communes Center for Experiential Learning (RC-CEL), United Nations International Children's Emergency Fund (UNICEF), Action for Community Empowerment (ACE), and the Himalayan Institute Hospital Trust (HIHT), for

assistance in facilitating the field research, particularly Satrabrata Acharyya (PRADAN), Avijit Mallik (PRADAN), Ramesh Abhishek (PRADAN), Mukesh Kumar (PRADAN), Malavika Chauhan (Tata Trusts), Vinod Kothari (Himmotthan Society), and Sunesh Sharma (Himmotthan Society). In addition, the authors thank Ramprasad Venkatesha (IIT-Bombay), Harshal Kate (IIT-Bombay), Aniket Deo (IIT-Bombay), Sanjay Kumar (ACE), Vivek Chauhan (UNICEF), Ganesh Bisht (Himmotthan Society), J. P. Belwal (HIHT), Nabajyoti Roy (PRADAN), Sambit Pradhan (PRADAN), and Prashant Kumar (PRADAN) for translation assistance during the interviews and workshops. Finally, the authors thank Shashank Deshpande (Maharashtra Groundwater Surveys Development Agency); Malini Shankar and S. C. Kollur (both Maharashtra Pollution Control Board); Karthik Seshan (Arghyam); Puneet Srivastava (WaterAid); and Prof. Manish Kumar (BIT-Mesra) for helpful meetings regarding water quality management, as well as the interview and workshop participants for this study.

## Author Contributions

**Conceptualization:** Michael S. Bono, Jr., Sydney Beasley, Emily Hanhauser, A. John Hart, Rohit Karnik, Chintan Vaishnav.

**Formal analysis:** Michael S. Bono, Jr.

**Funding acquisition:** A. John Hart, Rohit Karnik, Chintan Vaishnav.

**Investigation:** Michael S. Bono, Jr., Sydney Beasley, Emily Hanhauser, Rohit Karnik, Chintan Vaishnav.

**Methodology:** Michael S. Bono, Jr., Sydney Beasley, Chintan Vaishnav.

**Project administration:** Michael S. Bono, Jr., Chintan Vaishnav.

**Supervision:** Michael S. Bono, Jr., A. John Hart, Rohit Karnik, Chintan Vaishnav.

**Writing – original draft:** Michael S. Bono, Jr., Rohit Karnik, Chintan Vaishnav.

**Writing – review & editing:** Michael S. Bono, Jr., Sydney Beasley, Emily Hanhauser, A. John Hart, Rohit Karnik, Chintan Vaishnav.

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
