## [Decision Letter · Decision Letter 0]

21 Oct 2019

PONE-D-19-22906

Fieldwork-based Determination of Design Priorities for Point-of-Use Drinking Water Quality Sensors for Use in Resource-Limited Environments

PLOS ONE

Dear Dr. Bono Jr.,

Thank you for submitting your manuscript to PLOS ONE. After careful consideration, we feel that it has merit but does not fully meet PLOS ONE’s publication criteria as it currently stands. Therefore, we invite you to submit a revised version of the manuscript that addresses the points raised during the review process.

This is a very timely paper, with the potential to induce widespread impact for improving water quality and monitoring systems. 

I concur with the minor changes proposed by Reviewer 1.

 In addition, it would be useful if Figure 5 has a panel "d" added to it, reporting results of a conjoint analysis model pooled for all three populations shown in panels a, b and c of Figure 5. The discussion and conclusions section of the paper should clarify the differences observed among the three samples shown in panels a, b and c; and also compare these differences with the model estimated from pooling all samples together. 

Finally, the linkage of the study with "behavioral change communication" is not clear. How do the study findings from conjoint analysis induce behavioral change? What is the role of communication in inducing behavioral change, if any.

We would appreciate receiving your revised manuscript by Dec 05 2019 11:59PM. To enhance the reproducibility of your results, we recommend that if applicable you deposit your laboratory protocols in protocols.io, where a protocol can be assigned its own identifier (DOI) such that it can be cited independently in the future. For instructions see: http://journals.plos.org/plosone/s/submission-guidelines#loc-laboratory-protocols

We look forward to receiving your revised manuscript.

Kind regards,

Asim Zia, Ph.D.

Academic Editor

PLOS ONE

Journal Requirements:

2.  During our internal checks, the in-house editorial staff noted that you conducted research or obtained samples in another country. Please provide additional information, in the ethics statement in both the manuscript and submission information, regarding permits and approvals to conduct research in India. For instance, please discuss any permits or approvals obtained from the Indian Ministry of Home Affairs.

3. Please note that PLOS ONE does not have any length limitations, so consider moving information from the Supporting Information files to the main text of the paper.

'I have read the journal's policy and the authors of this manuscript have the following competing interests: The authors have submitted a utility patent application for potential sensor designs based on the results of this study: Bono MS, Beasley SB, Hanhauser EB, Vaishnav C, Hart AJ, Karnik RN. Systems, Devices, and Methods for Point-of-Use Testing for Fluid Contamination; Filed Oct. 18, 2017. U.S. PCT Patent Application No. PCT/US2017/057265.

At the time of article submission, none of the sensor designs described in this patent application have been commercialized, nor are there currently agreements in place to commercialize them.'

Additional Editor Comments (if provided):

This is a very timely paper, with the potential to induce widespread impact for improving water quality and monitoring systems.

I concur with the minor changes proposed by Reviewer 1.

In addition, it would be useful if Figure 5 has a panel "d" added to it, reporting results of a conjoint analysis model pooled for all three populations shown in panels a, b and c of Figure 5. The discussion and conclusions section of the paper should clarify the differences observed among the three samples shown in panels a, b and c; and also compare these differences with the model estimated from pooling all samples together.

Finally, the linkage of the study with "behavioral change communication" is not clear. How do the study findings from conjoint analysis induce behavioral change? What is the role of communication in inducing behavioral change, if any.

Reviewers' comments:

Reviewer's Responses to Questions

**Comments to the Author**

1. Is the manuscript technically sound, and do the data support the conclusions?

Reviewer #1: Yes

2. Has the statistical analysis been performed appropriately and rigorously? 

Reviewer #1: I Don't Know

3. Have the authors made all data underlying the findings in their manuscript fully available?

Reviewer #1: Yes

4. Is the manuscript presented in an intelligible fashion and written in standard English?

Reviewer #1: Yes

5. Review Comments to the Author

Reviewer #1: Review of fieldwork-based determination of design priorities for point-of-use drinking water quality sensors for use of in resource-limited environments

Bono et al. submitted to PLOS ONE

The authors address a profound challenge in the water quality monitoring field that has re-emerged with the sensor revolution in water quality monitoring. How to go about developing sensors that are useful/optimized for stakeholder applications? They focus on developing design priorities for point-of-use microbial water sensors for use in rural India households. I should note that I am a trained as a geochemist, and therefor am not qualified to assess the design and interpretations of their analysis of interactions with the stakeholder communities. I can say that the authors’ are asking the right questions and examining a useful combination of considerations that would go into development of sensors designed for use in various applied context. Therefore, my general assessment is that this is an extremely useful contribution to the water quality monitoring literature with broad and timely application to many subfields. We very much struggle with this ourselves in the nutrient monitoring world and these findings and approach have potential utility across water quality sensing; an indication of a strong contribution and manuscript worth publishing. I also found the manuscript particularly well-written and thoroughly proofread, which was refreshing and gives me minimal minor editorial comments that too often constitute much of a review. The authors should be commended for timely contribution and a well-written and structured manuscript. Based on my assessment, with the qualifier of my lack of expertise in this kind of stakeholder interaction experimental design and interpretation, I believe that the manuscript should be published with minimal revision. Some feedback on particular components of the manuscript that could be useful are mentioned below.

Table 1: This table is not particularly efficient use of space given that most of the cells are empty. Should be redesigned/condensed.

Figure 2 is hard to see in my document-check during prepublication review

Figure 3 caption: describe the example cards and how the choice is presented during the interview

Line 259-267: This is a peculiar way to introduce your Results and Discussion-as I read it, you are essentially presenting ‘hypotheses’ that your study does not test but rather using these hypotheses as justification for your study design and subsequent interpretation. If that is the intent, I would suggest reframing this to a more direct form of writing and eliminating ‘hypotheses’ in this section. E.g. Peer review literature suggests that there is a widespread need for…..Microbial contamination was most suitable for this study because….Furthermore, the most likely use case for improved…..

Line 341 and elsewhere: Your results indicate that not only output, but ‘recommended action’ is an important component of the monitoring approach. It would be useful to provide examples of how this may be done. E.g. details in the manual, color coding of measurements on sensor digital output based on concentration ranges, another data output indicative of action required etc. Perhaps this is another future line of research

Line 378(and elsewhere): It could be useful in follow up work to examine how the necessity or lack thereof in calibration requirements would impact these results. Many in-situ sensors require repeated calibration at various frequencies. It would seem to me that the frequency and complexity of calibrations would be an important control on the appeal of re-usable sensors to stakeholders as well. That is not addressed here, but important given your findings associated with a preference for reusable sensors. Perhaps something worth highlighting for future analyses.

6. PLOS authors have the option to publish the peer review history of their article (what does this mean?). If published, this will include your full peer review and any attached files.

Reviewer #1: No

---

## [Author Response · Author response to Decision Letter 0]

5 Jan 2020

We thank the reviewer and editorial staff for the constructive and helpful feedback. We have revised the manuscript accordingly, taking into account reviewer comments as well as editorial feedback. A manuscript with changes highlighted in blue is included with this submission. Our responses to comments appear below (and are also attached as a formatted document with this submission), with references as appropriate to line numbers in the revised manuscript.

Response to Reviewer #1 comments

• The authors address a profound challenge in the water quality monitoring field that has re-emerged with the sensor revolution in water quality monitoring. How to go about developing sensors that are useful/optimized for stakeholder applications? They focus on developing design priorities for point-of-use microbial water sensors for use in rural India households. I should note that I am a trained as a geochemist, and therefor am not qualified to assess the design and interpretations of their analysis of interactions with the stakeholder communities. I can say that the authors are asking the right questions and examining a useful combination of considerations that would go into development of sensors designed for use in various applied context. Therefore, my general assessment is that this is an extremely useful contribution to the water quality monitoring literature with broad and timely application to many subfields. We very much struggle with this ourselves in the nutrient monitoring world and these findings and approach have potential utility across water quality sensing; an indication of a strong contribution and manuscript worth publishing. I also found the manuscript particularly well-written and thoroughly proofread, which was refreshing and gives me minimal minor editorial comments that too often constitute much of a review. The authors should be commended for timely contribution and a well-written and structured manuscript. Based on my assessment, with the qualifier of my lack of expertise in this kind of stakeholder interaction experimental design and interpretation, I believe that the manuscript should be published with minimal revision. Some feedback on particular components of the manuscript that could be useful are mentioned below.

We thank the reviewer for their appreciative comments regarding the manuscript, and have endeavored to address their feedback for improvements as described below.

• Table 1: This table is not particularly efficient use of space given that most of the cells are empty. Should be redesigned/condensed.

We have made this change and redesigned this table for more efficient space usage.

• Figure 2 is hard to see in my document-check during prepublication review.

We have redesigned Figure 2 to make the text larger and easier to read.

• Figure 3 caption: describe the example cards and how the choice is presented during the interview

We have made this change, see Fig 3 caption and additional explanation in lines 310-314.

• Line 259-267: This is a peculiar way to introduce your Results and Discussion. As I read it, you are essentially presenting ‘hypotheses’ that your study does not test but rather using these hypotheses as justification for your study design and subsequent interpretation. If that is the intent, I would suggest reframing this to a more direct form of writing and eliminating ‘hypotheses’ in this section. E.g. Peer review literature suggests that there is a widespread need for…..Microbial contamination was most suitable for this study because….Furthermore, the most likely use case for improved…..

We have reframed this portion of the Results & Discussion section to present these hypotheses as assumptions which guided our experimental design, see lines 342-356.

• Line 341 and elsewhere: Your results indicate that not only output, but ‘recommended action’ is an important component of the monitoring approach. It would be useful to provide examples of how this may be done. E.g. details in the manual, color coding of measurements on sensor digital output based on concentration ranges, another data output indicative of action required etc. Perhaps this is another future line of research.

We have added additional explanation of how recommended actions might be implemented, see lines 478-484.

• Line 378 (and elsewhere): It could be useful in follow up work to examine how the necessity or lack thereof in calibration requirements would impact these results. Many in-situ sensors require repeated calibration at various frequencies. It would seem to me that the frequency and complexity of calibrations would be an important control on the appeal of re-usable sensors to stakeholders as well. That is not addressed here, but important given your findings associated with a preference for reusable sensors. Perhaps something worth highlighting for future analyses.

We have added additional discussion of how user responses to potential calibration steps could be explored in usability studies using functional prototypes, see lines 494-502.

 

Response to editorial comments

• It would be useful if Figure 5 has a panel "d" added to it, reporting results of a conjoint analysis model pooled for all three populations shown in panels a, b and c of Figure 5. The discussion and conclusions section of the paper should clarify the differences observed among the three samples shown in panels a, b and c; and also compare these differences with the model estimated from pooling all samples together. 

The pilot conjoint analysis data (shown in panel 5a) cannot be pooled with the full conjoint analysis data because the cost and ingredient levels are different between the two choice sets. We have pooled the full conjoint analysis data for both populations interviewed using the full conjoint analysis interview choice sets, corresponding to the main population and local contacts presented in panels 5b and 5c, respectively. We present this pooled data as panel d as requested, with additional discussion in lines 467-472.

• Finally, the linkage of the study with "behavioral change communication" is not clear. How do the study findings from conjoint analysis induce behavioral change? What is the role, if any, of communication in inducing behavioral change, if any?

We have added additional explanation linking our study with behavior change communication, see lines 26-32.

• Please note that PLOS ONE does not have any length limitations, so consider moving information from the Supporting Information files to the main text of the paper.

We have moved the discussion of study design for our design workshops and conjoint analysis interviews, as well as recommendations for interacting with local partner organizations, to the main text so that they will be more readily accessible. See in particular lines 186-225, 242-259, 280-301, and 567-643.

---

## [Editor Report · Decision Letter 1]

9 Jan 2020

Fieldwork-based Determination of Design Priorities for Point-of-Use Drinking Water Quality Sensors for Use in Resource-Limited Environments

PONE-D-19-22906R1

Dear Dr. Bono Jr.,

We are pleased to inform you that your manuscript has been judged scientifically suitable for publication and will be formally accepted for publication once it complies with all outstanding technical requirements.

With kind regards,

Asim Zia, Ph.D.

Academic Editor

PLOS ONE
---

## [Editor Report · Acceptance letter]

15 Jan 2020

PONE-D-19-22906R1 

Fieldwork-based Determination of Design Priorities for Point-of-Use Drinking Water Quality Sensors for Use in Resource-Limited Environments 

Dear Dr. Bono Jr.:

I am pleased to inform you that your manuscript has been deemed suitable for publication in PLOS ONE. Congratulations! Your manuscript is now with our production department. 

With kind regards,

on behalf of

Professor Asim Zia 

Academic Editor

PLOS ONE